# Factors Associated with and Prognosis Impact of Perceived Sleep Quality and Estimated Quantity in Patients Receiving Non-Invasive Ventilation for Acute Respiratory Failure

**DOI:** 10.3390/jcm11154620

**Published:** 2022-08-08

**Authors:** Matthieu Lê Dinh, Michael Darmon, Achille Kouatchet, Samir Jaber, Ferhat Meziani, Sebastien Perbet, Gerald Chanques, Elie Azoulay, Alexandre Demoule

**Affiliations:** 1Service de Médecine Intensive et Réanimation (Département R3S), AP-HP, Groupe Hospitalier Universitaire APHP-Sorbonne Université, Site Pitié-Salpêtrière, 75013 Paris, France; 2UMRS1158 Neurophysiologie Respiratoire Expérimentale et Clinique, INSERM, Sorbonne Université, 75005 Paris, France; 3Département d’Anesthésie—Réanimation, AP-HP, Groupe Hospitalier Universitaire APHP-Sorbonne Université, Site Pitié-Salpêtrière, 75013 Paris, France; 4Service de Médecine Intensive et Réanimation, Hôpital Saint-Louis, APHP and Université de Paris, 75019 Paris, France; 5Service de Réanimation Médicale et Médecine Hyperbare, Centre Hospitalier Régional Universitaire, 49100 Angers, France; 6Département d’Anesthésie et Réanimation, Hôpital Saint-Eloi, 34295 Montpellier, France; 7Montpellier School of Medicine, University of Montpellier, INSERM U1046, CNRS UMR 9214, 34093 Montpellier, France; 8Department of Intensive Care (Service de Médecine Intensive—Réanimation), Nouvel Hôpital Civil, Hôpital Universitaire de Strasbourg, 67000 Strasbourg, France; 9INSERM (French National Institute of Health and Medical Research), UMR 1260, Regenerative Nanomedicine (RNM), CRBS (Centre de Recherche en Biomédecine de Strasbourg), FMTS (Fédération de Médecine Translationnelle de Strasbourg), University of Strasbourg, 67000 Strasbourg, France; 10Réanimation Médico-Chirurgicale, CHU de Clermont-Ferrand, 63000 Clermont-Ferrand, France; 11R2D2, EA-7281, Université d’Auvergne, 63000 Clermont-Ferrand, France

**Keywords:** sleep, intensive care unit, acute respiratory failure, non-invasive ventilation

## Abstract

**Background.** The objectives of this study were (1) to determine factors associated with impaired sleep and (2) to evaluate the relationship between impaired sleep and the outcome. **Methods.** Secondary analysis of a prospective observational cohort study in 54 intensive care units in France and Belgium. Sleep quality was quantified by the patients with a semi-quantitative scale. **Results.** Among the 389 patients included, 40% reported poor sleep during the first night in the ICU and the median (interquartile) total sleep time was 4 h (2–5). Factors independently associated with poor sleep quality were the SOFA score (odds ratio [OR] 0.90, *p* = 0.037), anxiety (OR 0.43, *p* = 0.001) and the presence of air leaks (OR 0.52, *p* = 0.013). Factors independently associated with short-estimated sleep duration (<4 h) were the SOFA score (1.13, *p* = 0.005), dyspnea on admission (1.13, *p* = 0.031) and the presence of air leaks (1.92, *p* = 0.008). Non-invasive ventilation failure was independently associated with poor sleep quality (OR 3.02, *p* = 0.021) and short sleep duration (OR 0.77, *p* = 0.001). Sleep quality and duration were not associated with an increase in mortality or length of stay. **Conclusions.** The sleep of patients with ARF requiring NIV is impaired and is associated with a high rate of NIV failure.

## 1. Introduction

Patients receiving mechanical ventilation in the intensive care unit (ICU) exhibit severe sleep abnormalities [1]. These abnormalities include sleep deprivation and disruption as well as abnormal sleep architecture [2]. In turn, these sleep abnormalities may compromise weaning [3], affect a patient’s quality of sleep and life following hospital discharge [4] and may be associated with neurocognitive impairment [5].

To date, data on sleep in patients receiving non-invasive ventilation (NIV) in the ICU are scarce as most data is derived from studies conducted in intubated patients. However, the proportion of patients admitted for acute respiratory failure (ARF) and managed with NIV is increasing [6]. On the one hand, it is noticeable that NIV may improve sleep in patients with nocturnal alveolar hypoventilation [7,8,9,10]. On the other hand, patient ventilator asynchrony associated with NIV may alter sleep architecture [7,9]. Finally, altered sleep architecture is associated with late NIV failure and a higher incidence of delirium [11]. Most of these data come from studies that included a small sample size but evaluated sleep with high quality physiological tools such as polysomnography. In addition, these studies focused on the specific population of patients with chronic obstructive pulmonary disease admitted in the ICU for an acute severe exacerbation.

The primary objective of this study was to evaluate perceived sleep quality and estimated total sleep time during the first night following ICU admission in a large population of patients receiving NIV for ARF of various etiologies. We also examined factors associated with these outcomes. Finally, we investigated the impact of sleep quality and quantity on NIV success or failure, on length of stay and mortality and on post-ICU burden.

## 2. Materials and Methods

This is a secondary analysis of a prospective observational study conducted in 54 French and Belgian ICUs members of the REVA Network (Research Network in Mechanical Ventilation) or the FAMIREA Study Group (on improving the effectiveness of communication with the relatives of ICU patients). The parent study was designed to evaluate NIV use in terms of frequency, indications, outcome and ICU burden. We have listed participating centres and collaborators in the acknowledgements section. The study was approved by the institutional review board of the Société de Pneumologie de Langue Française n. CEPRO 2010-015 and was registered on a publicly available database (ClinicalTrials.gov, identifier: NCT01449331). We obtained written informed consent from all patients or their relatives. Three other studies based on this cohort have been already published elsewhere [6,12,13].

### 2.1. Study Population

Over a 2-month enrolment period between November 2010 and April 2011, each participating ICU included in the prospective parent cohort study consecutive adults requiring ventilatory assistance (either invasive mechanical ventilation or NIV) for ARF. Acute respiratory failure was defined by a respiratory rate of >30 cycles·min^−1^, signs of respiratory distress, or arterial oxygen saturation measured by pulse oximetry (SpO_2_) < 90% on room air. In the present post hoc study, we only included patients who received NIV as a first-line treatment for ARF. Patients receiving NIV for comfort care only were excluded. Patients with missing data or incomplete data on sleep quality were also excluded.

### 2.2. Data Collection

We followed patients daily in the ICU, at hospital discharge and at 90 days post-ICU discharge (day 90). The demographic data and medical history collected consisted of age, gender, Simplified Acute Physiology Score (SAPS) II [14], Sequential Organ Failure Assessment score (SOFA) [15] and underlying diseases such as chronic respiratory disease, chronic heart failure and immunosuppression. The cause of ARF was classified as either acute-on-chronic respiratory failure (acute respiratory failure occurring in patients with pre-existing respiratory disease), cardiogenic pulmonary oedema, or de novo ARF (defined as respiratory failure not exacerbating chronic lung disease or heart failure, also called hypoxemic ARF). Respiratory rate, intensity of dyspnea (assessed with a modified Borg scale) and arterial blood gas values were recorded first at ICU admission before the initiation of ventilatory support, and second after the first NIV session.

To assess the quality of sleep, nurses asked the patient to rate perceived quality of sleep following the first night in the ICU [16]. This was carried out on a scale that consisted of verbal descriptors linked to specific numbers. This scale ranges from 1 to 4, representing a very poor, acceptable, good and very good sleep quality, respectively. It was used to identify two clinically relevant groups of patients based on sleep quality. A sleep quality of 1 was defined as “poor sleep quality”, while a sleep quality of 2 or more was defined as “acceptable to very good sleep quality”. Total sleep duration during the first night following admission was estimated by nurses. Patients were separated into two groups according to median estimated sleep time duration (4 h). Short sleep duration was defined as estimated sleep time <4 h during the first 24 h following inclusion. Acceptable sleep duration was defined by estimated sleep time ≥4 h. The presence of air leaks and anxiety was recorded. We also recorded the need for invasive mechanical ventilation, ICU and in-hospital length of stay, ICU and in-hospital mortality and day 90 mortality. NIV success or failure was defined as follows: Patients requiring endotracheal intubation or died during the 24 h following NIV discontinuation were classified as NIV failures. Patients treated with NIV until they no longer required ventilatory support were classified as NIV successes. Ninety days after ICU discharge (day 90), trained social workers coached by psychologists and sociologists of the FAMIREA study group interviewed survivors by telephone. Patients were asked to complete the Impact of Event Scale-Revised (IES-R) to assess post-traumatic stress disorder related symptoms [17], and the Hospital Anxiety and Depression Scale (HADS) to quantify symptoms of anxiety and depression [18] (in that order). Lower HADS and IES-R scores indicated less post-ICU burden.

### 2.3. Data Quality

An ICU physician not involved in the study resolved inconsistencies in the data entered by the investigators, based on comparison of the study case report forms with the medical charts. The database was audited by an independent check of all ICU variables on a random sample of 10% of patients.

### 2.4. Statistical Analysis

Quantitative variables were described as median (interquartile range (IQR)) and were compared between groups using the non-parametric Wilcoxon rank-sum test. Qualitative variables were described as n (%) and were compared between groups using Fisher’s exact test. Correlation between perceived sleep quality and estimated total sleep time was estimated and tested using Spearman’s correlation coefficient with its 95% confidence interval (95% CI). Estimated sleep duration according to sleep quality was assessed with a Kruskal–Wallis test, followed, if significant, by a Dunn test. Factors associated with poor sleep quality, short sleep duration (<4 h) and NIV failure were studied by multivariate logistic regression. Multivariate models were built with variables that yielded *p*-values less than 0.2 on univariate analysis and/or were considered to be clinically relevant. Sleep duration was not included in the multivariate analysis model exploring sleep quality and vice versa because of their correlation. Likewise, two multivariate analysis models were performed for associated factors with NIV failure, one with sleep quality, the other with sleep duration. Adjusted odds ratios (ORs) of variables present in the final multiple logistic regression models are presented with their 95% CIs. Linear regression coefficients of variables present in the final multiple linear regression model (sleep duration) are presented with their standard error. All tests were two-sided and *p*-values less than 0.05 were considered statistically significant. Statistics were performed with the R software (version 3.5.2, R Foundation for Statistical Computing, Vienna, Austria, https://www.R-project.org/).

## 3. Results

### 3.1. Study Population and Quality of Sleep

During the study period, 2367 patients requiring ventilatory support were admitted to the ICU (Figure 1).

Of these, 1799 received invasive ventilatory support, 1203 for a non-respiratory condition and 596 as a first-line treatment for ARF. The remaining 568 patients received NIV as a first-line treatment for ARF. Of these, 61 patients who received comfort-only NIV and 118 patients with missing or incomplete data on sleep quality were excluded from the study. A total of 389 patients were finally assessed for sleep quality and were included in the present study. Among them, 320 were assessed for sleep duration. Table 1 describes patient characteristics.

During the first 24 h following inclusion, the median sleep quality score was 2 (1–2). It was ≥2 (acceptable to very good sleep quality) in 234 patients (60%). The median total sleep time was 4 h (2–5). There was a positive correlation between sleep quality and estimated total sleep time (Rho = 0.73; 95% CI 0.67–0.78; *p* < 0.0001) (Figure 2).

### 3.2. Factors Associated with Poor Sleep Quality and Short Sleep Duration

Table 1 displays the factors associated with poor perceived sleep quality during the first night following admission in the ICU. On multivariate logistic regression analysis and after the selection process, three factors were independently associated with poor sleep quality: the SOFA score, anxiety and the presence of air leaks.

Table 2 displays the factors associated with short-estimated sleep duration (<4 h) during the first night following admission in the ICU. On multivariate logistic regression analysis and after the selection process, three factors were independently associated with short sleep duration (<4 h): the SOFA score, dyspnea on admission and the presence of air leaks.

### 3.3. Factors Associated with NIV Failure

NIV failure rate was 19.8% (n = 77). Table 3 displays the factors associated with NIV failure or success by univariate analysis and Table 4 displays the factors associated with NIV failure or success by multivariate analysis. On the multivariate analysis model including perceived sleep quality and after the selection process, three factors independently predicted the result of NIV: PaCO_2_ on admission was associated with NIV success, while poor sleep and the SOFA score were associated with NIV failure. On the multivariate analysis model including sleep duration and after the selection process, four factors independently predicted the result of NIV: PaCO_2_ on admission and sleep duration were associated with NIV success, while de novo ARF and SOFA score were associated with NIV failure.

Sleep duration was not included in the multivariate analysis model exploring sleep quality and vice versa because of their correlation. Likewise, two multivariate analysis models were performed for associated factors with NIV failure, one with sleep quality, the other with sleep duration.

### 3.4. Association between Sleep Quality or Duration and Outcome and Post ICU Burden

Appendix A in the Online Supplement displays mortality, length of stay, quality of life and post-ICU burden. Poor sleep quality and short-estimated sleep time were not associated with any alteration of mortality or length of stay. HADS anxiety and depression subscores and IES-R did not indicate a greater burden in patients with poor perceived sleep quality or short-estimated sleep time.

## 4. Discussion

The main and major findings of this study can be summarized as follows; in a population of patients admitted to the ICU for ARF requiring NIV: (1) the perceived sleep quality is poor in 40% of the patients and estimated total sleep duration is short, and (2) poor perceived sleep quality and short-estimated sleep duration are independently associated with NIV failure.

To the best of our knowledge, this is the largest study to investigate sleep in a population of non-intubated patients admitted for ARF and treated with NIV. In the ICU, sleep has been mostly studied in intubated patients [2,19,20] or in those treated with NIV for acute-on-chronic respiratory failure [11] or neuromuscular diseases [7,8]. However, factors associated with sleep quality and duration and the relationship between sleep and the prognosis have not been previously studied in such a large population.

### 4.1. Sleep Quality and Duration

The perceived sleep quality was globally low, with a median score of 2 on a semi-quantitative scale range from 1 to 4, which agrees with previous published data. On the other hand, estimated total sleep time was 4 h in our study. This result was under the 6 to 9 h found in previous studies conducted in patient under invasive mechanical ventilation [2]. It was, however, close to the 5.8 h median duration of sleep observed by Roche-Campo in patients with acute-on-chronic ARF treated with NIV [11]. Of note, sleep was not assessed at the same time point in the two studies; it was assessed between the second and the fourth night in the study by Roche-Campo et al., whereas it was assessed within the 24 h following ICU admission in our study [11]. It is important to point that the impact of NIV on sleep is not unique. On the one hand, NIV may induce sleep disruption due to interface intolerance and patient-ventilator asynchrony [21]. On the other hand, NIV may improve sleep quality by decreasing inspiratory effort and obstructive sleep apnea [22].

### 4.2. Factors Associated with Low Sleep Duration and Quality

Our results identified factors associated with poorer and shorter sleep. Among them, some had been previously evidenced, such as patient severity, which is associated with poorer sleep in the coronary care unit and in the ICU [23,24,25]. Air leaks is an established factor of altered sleep in patients receiving home NIV [26]. In the ICU patients receiving NIV, leaks contribute to patient ventilator asynchrony [27], which in turn alters sleep quality and architecture [28,29]. Anxiety and sleep have a bidirectional relationship. Anxiety causes sleep deprivation [30,31] and sleep deprivation induces a state of increased anxiety [32]. Finally, we showed for the first time in the ICU a statistical link between dyspnea and low sleep duration. In patients with chronic breathlessness and in cancer patients, there is a strong association between dyspnea and sleep problems [33,34]. The mechanism that links dyspnea and sleep problems is unclear. It could be mediated by anxiety, since anxiety and dyspnea are strongly linked [13,35]. It could also involve abnormal afferent messages from neck inspiratory muscles, which are often recruited in case of dyspnea and ARF [36].

### 4.3. Relationship between Sleep and NIV Failure

We found that poor sleep quality and short sleep duration were associated with a higher rate of NIV failure and subsequent intubation. Because this independent association was adjusted on severity (SOFA score), it suggests that the link between poor sleep and NIV failure is not exclusively explained by the fact that the sickest patients are both at a higher risk of poor sleep and NIV failure. A relationship between sleep abnormalities and late NIV failure has been established in patients with an acute hypercapnic respiratory failure [11]. In this study, patients who failed NIV had more abnormal sleep patterns, a lower night-to-day total sleep time ratio and a lower duration of REM sleep. Sleep abnormalities are also known to be associated with delirium [11]. They could reflect subclinical manifestations of an acute brain dysfunction that may directly influence the outcome of NIV. Indeed, sleep deprivation reduces respiratory motor output by altering its cortical component with a subsequent reduction in inspiratory endurance [37].

### 4.4. Limitations

This study presents limitations that need to be acknowledged. First, sleep was assessed in a subjective way. Although polysomnography is the reference method, it remains difficult to perform to and interpret properly in the ICU setting. The use of a subjective method was more feasible. Despite a tendency to overestimate quality and duration of sleep, there is a correlation between subjective assessment and polysomnography [38,39]. In addition, PADIS guidelines suggest to not routinely use physiologic sleep monitoring in the ICU and to prefer the monitoring of patients perceived sleep [16]. Second, this study is a post hoc analysis of a prospective cohort of patients with ARF, which the primary objective was not to study sleep and that it was conducted 10 years ago. Given the improvement in some technologies, it might not reflect the current situation. However, despite the improvement in the performance of ventilators, leaks, for instance, are still an issue in some patients. Third, the presence of a delirium, poor quality of sleep prior to ICU, level 3 analgesics or sedative anxiolytics use were not collected and could be confounding factors by participating simultaneously in sleep alteration and NIV failure. Fourth, sleep duration was assessed by nurses, as opposed to sleep quality, which was self-reported. In ICU patients, it is well demonstrated that abnormal pattern, such as atypical sleep and pathological wakefulness, are common and can be misleading [40]. Indeed, a patient with open eyes can be asleep and a patient with closed eyes can be awake. Fifth, we only measured sleep quality and quantity once, on the first night following admission. We cannot exclude the fact that sleep quality and quantity were different on the second or third night as compared to the first night. Sixth, the absence of a link between poor sleep and post-ICU burden should be taken cautiously. Indeed, sleep quality was assessed only during the first night and we cannot rule out that this is the repetition of poor sleep associated with post-ICU burden. In addition, due to the high proportion of patients lost to follow up, ICU burden was evaluated on a small sample.

## 5. Conclusions

In conclusion, the sleep of patients with ARF requiring NIV is impaired regardless of the aetiology of ARF. Poor sleep is definitely an ICU stressor, among others such as anxiety, pain, thirst and dyspnea. Subsequently, poor sleep requires the attention of stakeholders, such as pain. Improving the treatment of poor sleep in order to improve patient comfort deserves evaluation. Future work may compare various strategies to improve sleep in ICU patients and assess the impact of these strategies on short-term outcomes such as NIV failure, but also on middle- and long-term outcomes such as the ICU burden.

Sleep could also be used as a monitoring tool to anticipate NIV failure and, subsequently, to avoid delaying intubation too much. Future works might assess the benefit of routine sleep monitoring in patients admitted for ARF. Future work might also assess the potential benefit of pharmacological treatment, aiming to improve sleep and perceived sleep quality and reduce the chance of NIV failure.

## Figures and Tables

**Figure 1 jcm-11-04620-f001:**
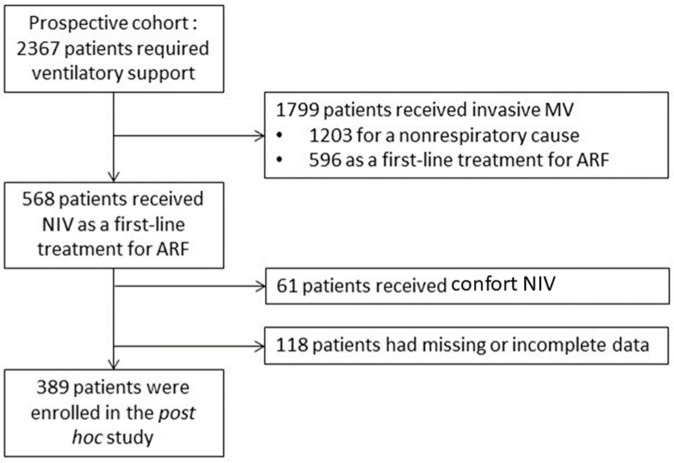
Flow chart of the study. MV, mechanical ventilation; ARF, acute respiratory failure; NIV, non-invasive ventilation.

**Figure 2 jcm-11-04620-f002:**
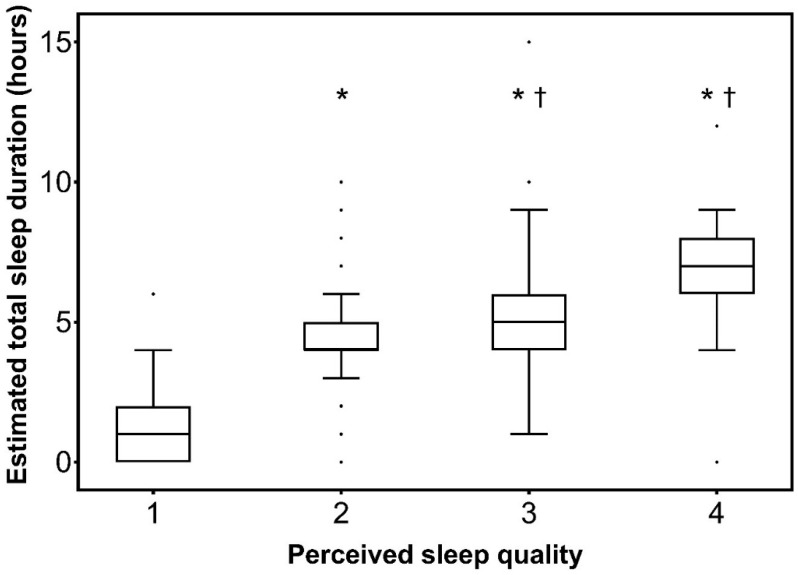
Estimated total sleep duration according to sleep quality. Semi-quantitative sleep quality scale: 1, very poor; 2, acceptable; 3, good; 4, very good. Results are presented in the form of box plot. Boxes are drawn between the first and third quartiles of the distribution, black bars indicate the median, whiskers are made with Tukey method, and dots are the outliers. * *p* < 0.0001 vs. 1; ^†^ *p* < 0.05 vs. 2.

**Table 1 jcm-11-04620-t001:** Univariate and multivariate analysis: factors associated with poor perceived sleep quality.

	Univariate Analysis				Multivariate Analysis	
	All Patients (*n* = 389)	Poor Sleep Quality (*n* = 155)	Acceptable-to-Very Good Sleep Quality (*n* = 234)	*p*-Value	Odds Ratio (95% Confidence Interval)	*p*-Value
**Patient characteristics**						
Age, *years*	69 (59–77)	67 (59–76)	69 (59–78)	0.307		
Males, *n* (%)	248 (64)	99 (64)	149 (64)	1.000		
BMI, kg·m^−2^	26 (22–33)	26 (22–31)	26 (23–33)	0.553		
Chronic respiratory disease, *n* (%)	249 (64)	98 (63)	151 (65)	0.830		
Chronic cardiac disease, *n* (%)	86 (22)	36 (23)	50 (21)	0.709		
Home oxygen therapy, *n* (%)	80 (21)	26 (17)	54 (23)	0.158		
**NIV episode**						
SAPS 2	35 (27–44)	36 (28–46)	34 (26–43)	0.065		
SOFA	3 (2–5)	3 (2–6)	3 (2–5)	0.278	1.11 (1.01–1.23)	0.037
Cause of ARF						
Acute-on-chronic, *n* (%)	234 (60)	88 (57)	146 (62)	0.336		
Acute cardiogenic pulmonary edema, *n* (%)	107 (28)	49 (32)	58 (25)			
De novo ARF, *n* (%)	48 (12)	18 (12)	30 (13)			
**On ICU admission, prior to NIV**						
Respiratory rate, *cycle*·min^−1^	32 (28–37)	32 (28–37)	31 (26-36)	0.361		
Dyspnea Borg scale	4 (3–5)	4 (3–5)	4 (3–5)	0.194		
*Blood gases*						
PaO_2_/FiO_2_, mmHg	223 (161–287)	222 (157–273)	223 (163–298)	0.356		
PaCO_2_, mmHg	53 (40–71)	53 (36–69)	54 (42–72)	0.170		
pH	7.33 (7.27–7.40)	7.34 (7.27–7.42)	7.33 (7.27–7.40)	0.737		
**After the first NIV session**						
Air leaks, *n* (%)	237 (61)	107 (69)	130 (56)	0.008	1.92 (1.15–3.23)	0.013
Anxiety, *n* (%)	216 (56)	98 (63)	118 (50)	<0.001	2.33 (1.39–3.85)	0.001
Respiratory rate, *cycle*·min^−1^	27 (23–33)	29 (24–35)	26 (23–30)	0.005		
Dyspnea Borg scale	3 (2–4)	4 (3–5)	3 (2–4)	<0.001		
*Blood gases*						
PaO_2_/FiO_2_, mmHg	213 (163–257)	193 (150–240)	220 (174–283)	0.083		
PaCO_2_, mmHg	54 (44–66)	54 (41–65)	54 (42–68)	0.868		
pH	7.35 (7.29–7.40)	7.35 (7.29–7.40)	7.35 (7.29–7.40)	0.803		
NIV interface				0.282		
Oro-nasal mask	245 (63)	94 (61)	151 (65)			
Nasal mask	4 (1)	1 (1)	3 (1)			
Full face mask	64 (16)	31 (20)	33 (14)			
Type of ventilator				0.192		
NIV dedicated ventilator, *n* (%)	51 (13)	16 (10)	35 (15)			
ICU ventilator, *n* (%)	167 (43)	71 (46)	96 (41)			
Total sleep duration over the first 24 h after admission, h	4 (2–5)	1 (0–2)	5 (4–6)	<0.001		

For univariate analysis, data are presented as n (%) or median (interquartile range). BMI, body mass index; NIV, non-invasive ventilation; SAPS2, Simplified Acute Physiology Score 2; SOFA, Sequential Organ Failure Assessment score; ARF, acute respiratory failure; ICU, intensive care unit; PaO_2_, arterial oxygen tension; FiO_2_, inspiratory oxygen fraction; PaCO_2_, arterial carbon dioxide tension. The following variables were included into the initial complete multivariate model: home oxygen therapy, SOFA (rather than SAPS2, the two variables being colinear), dyspnea Borg scale on ICU admission, PaCO_2_ on ICU admission, air leaks, anxiety after the first NIV session, respiratory rate after the first NIV session, PaO_2_/FiO_2_ after the first NIV session and type of ventilator.

**Table 2 jcm-11-04620-t002:** Univariate and multivariate analysis: factors associated with short (<4 h) estimated sleep duration.

		Univariate Analysis		Multivariate Analysis	
	Short Sleep Duration < 4 h (*n* = 148)	Acceptable Sleep Duration ≥ 4 h (*n* = 172)	*p*-Value	Odds Ratio (95% Confidence Interval)	*p*-Value
**Patient characteristics**					
Age, *years*	70 (57–7)	70 (61–80)	0.071		
Males, *n* (%)	97 (66)	108 (63)	0.641		
BMI, kg.m^−2^	26 (22–31)	26 (23–32)	0.488		
Chronic respiratory disease, *n* (%)	92 (62)	121 (70)	0.125		
Chronic cardiac disease, *n* (%)	31 (21)	40 (23)	0.686		
Home oxygen therapy, *n* (%)	20 (14)	46 (27)	0.004		
**NIV episode**					
SAPS 2	40 (29–49)	34 (27–42)	0.001		
SOFA	4 (2–7)	3 (2–5)	0.011	1.13 (1.04–1.23)	0.005
*Cause of ARF*					
Acute-on-chronic, *n* (%)	81 (55)	112 (65)	0.082		
Acute cardiogenic pulmonary edema, *n* (%)	48 (32)	37 (22)			
De novo ARF, *n* (%)	19 (13)	23 (13)			
**On ICU admission, prior to NIV**					
Respiratory rate, *cycle*·min^−1^	32 (28–40)	30 (25–36)	0.016		
Dyspnea Borg scale	4 (3–5)	3 (3–4)	0.013	1.13 (1.01–1.27)	0.031
*Blood gases*					
PaO_2_/FiO_2_, mmHg	209 (141–269)	223 (163–292)	0.129		
PaCO_2_, mmHg	52 (36–70)	56 (43–72)	0.041		
pH	7.33 (7.25–7.41)	7.33 (7.28–7.40)	0.649		
**After the first NIV session**					
Air leaks, *n* (%)	103 (70)	97 (56)	0.016	1.92 (1.18–3.14)	0.008
Anxiety, *n* (%)	92 (62)	87 (51)	0.022		
Respiratory rate, *cycle*·min^−1^	30 (24–35)	25 (22–30)	<0.001		
Dyspnea Borg scale	4 (3–5)	3 (2–3)	<0.001		
*Blood gases*					
PaO_2_/FiO_2_, mmHg	205 (150–250)	222 (184–283)	0.114		
PaCO_2_, mmHg	53 (41–64)	56 (43–68)	0.400		
pH	7.35 (7.27–7.40)	7.36 (7.31–7.40)	0.511		
*NIV interface*			0.089		
Oro-nasal mask	106 (72)	104 (60)			
Nasal mask	0 (0)	2 (1)			
Full face mask	18 (12)	31 (18)			
*Type of ventilator*			0.457		
NIV dedicated ventilator, *n* (%)	15 (10)	21 (12)			
ICU ventilator, *n* (%)	72 (49)	72 (42)			
Very poor perceived quality of sleep over the first 24 h after admission, *n (%)*	113 (76)	10 (6)	<0.001		

Data are presented as n (%) or median (interquartile range); short sleep duration is defined as an estimated sleep time < 4 h while acceptable sleep duration is defined as an estimated sleep time ≥ 4 h. BMI, body mass index; NIV, non-invasive ventilation; SAPS2, Simplified Acute Physiology Score 2; SOFA, Sequential Organ Failure Assessment score; ARF, acute respiratory failure; ICU, intensive care unit; PaO_2_, arterial oxygen tension; FiO_2_, inspiratory oxygen fraction; PaCO_2_, arterial carbon dioxide tension. The following variables were included into the initial complete multivariate model: age, chronic respiratory disease, SOFA (rather than SAPS 2, the two variables being colinear), cause of acute respiratory failure, respiratory rate on ICU admission, dyspnea Borg scale on ICU admission, PaO_2_/FiO_2_ on ICU admission, PaCO_2_ on ICU admission, air leaks, anxiety after the first NIV session, respiratory rate after the first NIV session, dyspnea Borg scale after the first NIV session, PaO_2_/FiO_2_ after the first NIV session and NIV interface.

**Table 3 jcm-11-04620-t003:** Univariate analysis: factors associated with NIV failure.

	NIV Failure (*n* = 77)	NIV Success (*n* = 312)	*p*-Value
**Patient characteristics**			
Age, *years*	66 (57–76)	69 (59–78)	0.108
Males, *n* (%)	55 (71)	193 (62)	0.145
BMI, kg.m^−2^	26 (23–30)	26 (22–33)	0.857
Chronic respiratory disease, *n* (%)	39 (51)	210 (67)	0.008
Chronic cardiac disease, *n* (%)	14 (18)	72 (23)	0.363
Home oxygen therapy, *n* (%)	9 (12)	71 (23)	0.028
**NIV episode**			
SAPS 2	44 (34–57)	33 (26–42)	<0.001
SOFA	6 (3–9)	3 (2–4)	<0.001
*Cause of ARF*			<0.001
Acute-on-chronic, *n* (%)	32 (42)	202 (65)	
Acute cardiogenic pulmonary edema, *n* (%)	38 (49)	69 (22)	
*De novo* ARF, *n* (%)	7 (9)	41 (13)	
On ICU admission, prior to NIV			
Respiratory rate, *cycle*·min^−1^	34 (28–40)	31 (27–36)	0.042
Dyspnea Borg scale	4 (3–5)	3 (2–5)	0.001
*Blood gases*			
PaO_2_/FiO_2_, mmHg	176 (118–230)	232 (180–295)	<0.001
PaCO_2_, mmHg	43 (32–54)	60 (43–72)	<0.001
pH	7.36 (7.28–7.44)	7.33 (7.27–7.40)	0.121
**After the firt NIV session**			
Leaks, *n* (%)	48 (62)	189 (61)	0.796
anxiety, *n* (%)	51 (74)	165 (59)	0.027
Respiratory rate, *cycle*·min^−1^	30 (24–36)	27 (23–32)	0.023
Dyspnea Borg scale	4 (3–5)	3 (2–5)	0.014
*Blood gases*			
PaO_2_/FiO_2_, mmHg	170 (143–214)	222 (177–283)	0.002
PaCO_2_, mmHg	48 (34–57)	57 (45–69)	<0.001
pH	7.35 (7.26–7.41)	7.35 (7.29–7.40)	0.516
*NIV interfaces*			0.178
Oro-nasal mask	52 (68)	193 (62)	
Nasal mask	0 (0)	4 (1)	
Full face mask	8 (10)	56 (18)	
*Type of ventilator*			0.003
NIV dedicated ventilator, *n* (%)	3 (4)	48 (15)	
ICU ventilator, *n* (%)	42 (55)	125 (40)	
**First 24 h after admission**			
Poor perceived quality of sleep, *n* (%)	46 (60)	109 (35)	<0.001
Total sleep time, h	2 (0–4)	4 (2–6)	<0.001

Data are presented as *n* (%) or median (interquartile range), unless otherwise stated. BMI, body mass index; SAPS, Simplified Acute Physiology Score; SOFA, Sequential Failure Assessment score; ARF, acute respiratory failure; NIV, non-invasive ventilation; PaO_2_, arterial oxygen tension; FiO_2_, inspiratory oxygen fraction; PaCO_2_, arterial carbon dioxide tension.

**Table 4 jcm-11-04620-t004:** Multivariate analysis: factors associated with NIV failure.

	Sleep Quality		Sleep Quantity	
	Odds Ratio (95% Confidence Interval)	*p*-Value	Odds Ratio (95% Confidence Interval)	*p*-Value
SOFA	1.26 (1.14–1.41)	0.0001	1.33 (1.20–1.49)	0.0001
Cause of ARF, de novo ARF, *n* (%)			3.21 (1.11–10.6)	0.039
PaCO_2_ on ICU admission, prior to NIV, *per* mmHg	0.98 (0.96–0.99)	0.006	0.98 (0.96–0.99)	0.003
Poor perceived quality of sleep	3.02 (1.26–8.49)	0.021		
Total sleep time, *per* hour			0.77 (0.66–0.88)	0.001

SOFA, Sequential Failure Assessment score; ARF, acute respiratory failure; PaCO_2_, arterial carbon dioxide tension; NIV, non-invasive ventilation. The following variables were included into the initial complete multivariate model: chronic respiratory disease, SOFA (rather than SAPS 2, the two variables being colinear), cause of acute respiratory failure, respiratory rate on ICU admission, dyspnea Borg scale on ICU admission, PaO_2_/FiO_2_ on ICU admission, PaCO_2_ on ICU admission, anxiety after the first NIV session.

## Data Availability

The datasets used and/or analysed during the current study are available from the corresponding author on reasonable request.

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
