# Peer review of "Factors Associated with and Prognosis Impact of Perceived Sleep Quality and Estimated Quantity in Patients Receiving Non-Invasive Ventilation for Acute Respiratory Failure"

_jcm, 2022, doi:10.3390/jcm11154620_

Round 1

Reviewer 1 Report

Thank you for giving me the opportunity to check the article. 
This article is described the relationship between the quality of sleep and acute respiratory failure in ICU as a prognostic factor. It's very interesting for intesivists and physicians. I agree with your revised article without any comments.

Author Response

Reviewer

Thank you for giving me the opportunity to check the article. This article is described the relationship between the quality of sleep and acute respiratory failure in ICU as a prognostic factor. It's very interesting for intesivists and physicians. I agree with your revised article without any comments.

Answer

We are grateful to the reviewer for her/his positive appraisal of our manuscript.

Reviewer 2 Report

I read with great interest the recent work of Le Dinh et al. investigating the sleep of 389 patients receiving non-invasive ventilation for acute respiratory failure in an ICU surrounding. I really congratulate the authors on their very informative work, because the group of non-invasive respiratory treated patients has scientifically been neglected so far.  Poor sleep can adversely affect patients’ psychological well-being. Patients rank sleep disruptions as one of the main causes of ICU-related distress. Poor sleep quality is not only a significant cause of emotional distress but also contributes to cognitive dysfunction, ICU delirium, impaired immune function, and prolonged mechanical ventilation, as also mentioned in the article. The structure of the paper is very well presented, the data are clear and understandable. All the limitations, that I noted while reading, were mentioned in the end. Maybe I could encourage the authors, to add their suggestions how sleep could be pharmacologically be improved regarding the described factors anxiety and dyspnea while trying to avoid a drug-induced delirium of patients at risk and not masking the sleep disturbance as possible monitoring tool to anticipate NIV failure in their conclusions. All in all, I would recommend acceptance after minor corrections of the spelling: e.g., “night” instead of “nigh”, page 1 line 73, “comfort” instead of “confort”, page 5 figure 1, or “adjusted” instead of “adjuted”, page 14 line 332.

Author Response

1)

Reviewer

I read with great interest the recent work of Le Dinh et al. investigating the sleep of 389 patients receiving non-invasive ventilation for acute respiratory failure in an ICU surrounding. I really congratulate the authors on their very informative work, because the group of non-invasive respiratory treated patients has scientifically been neglected so far.  Poor sleep can adversely affect patients’ psychological well-being. Patients rank sleep disruptions as one of the main causes of ICU-related distress. Poor sleep quality is not only a significant cause of emotional distress but also contributes to cognitive dysfunction, ICU delirium, impaired immune function, and prolonged mechanical ventilation, as also mentioned in the article. The structure of the paper is very well presented, the data are clear and understandable. All the limitations, that I noted while reading, were mentioned in the end.

Answer

We are grateful to the reviewer for her/his positive appraisal of our manuscript. We have taken all of the reviewer remarks into account, and have carefully revised the manuscript in line with her/his suggestions, as described in our point-by-point response. All changes made to the manuscript are indicated in red in the revised version.

2)

Reviewer

Maybe I could encourage the authors, to add their suggestions how sleep could be pharmacologically be improved regarding the described factors anxiety and dyspnea while trying to avoid a drug-induced delirium of patients at risk and not masking the sleep disturbance as possible monitoring tool to anticipate NIV failure in their conclusions.

Answer

We thank the reviewer for her/his suggestion. In the reviesed version of the manuscript, we have added the following sentence to the conclusion: “Future work might also assess the potential benefit of pharmacological treatment aiming at improving sleep of perceived sleep quality and on NIV failure”.

3)

Reviewer

All in all, I would recommend acceptance after minor corrections of the spelling: e.g., “night” instead of “nigh”, page 1 line 73, “comfort” instead of “confort”, page 5 figure 1, or “adjusted” instead of “adjuted”, page 14 line 332

Answer

We thank the reviewer for pointing these typos that we have corrected.

This manuscript is a resubmission of an earlier submission. The following is a list of the peer review reports and author responses from that submission.

Round 1

Reviewer 1 Report

Thank you for giving me the opportunity to check your article. This article described about the quality of sleep under respiratory support with non-invasive ventilation for acute respiratory failure in ICU. It's a very interesting presentation I think. Under ICU, in most of patients, altered sleep architecture is associated with late NIV failure and a higher incidence of delirium. It induced poor prognosis to them. So, your analysis is valuable.

But I have some questions to you.

  1. In study populations, this study covers cases between November 2010 and April 2011, but more than 10 years have passed since the study was conducted, and the development of medical technology, non-invasive respiratory technology, and drugs, so I don't think that it reflects the current situation, but what do you think.
  2. In data collection, you assessed the quality of sleep with interview using with defined score by nurses  following the first night in the ICU. But Is it acceptable to assess sleep quality only on the first day of ICU admission? I think it's difficult to judge from the quality of sleep on the first day alone, because daily sleep disorders in the ICU may cause delirium. What do you think about it? What is your opinion?
  3. Is it possible to make a proper evaluation by subjectively evaluating the defined factors associated with this interview with patients, because there are significant differences in the values of individual patients? We think that the evaluation with more objective data such as electroencephalography and capnometers is more accurate. What do you think about it?
  4. In result and discussion, you mentioned that air leak and anxiety under NIV were related to quality of sleep with significant difference. Do you have any solution of this problem? Under mechanical ventilation, in general, we use sedation drugs like fentanyl, propofol or others to manage sedation during ventilator management. However, under NIV, I don't think there are clear rules for sedation, have you taken any measures against this? Or do you take any steps to improve quality of in response to this result, for example, using with dexmedetomidine or psychiatric drugs? 

Please answer these clinical questions. 

Author Response

We thank the Reviewer for her/his careful reading and positive appraisal of our manuscript and for his comments and suggestions that we have taken into account.

1)

Reviewer

In study populations, this study covers cases between November 2010 and April 2011, but more than 10 years have passed since the study was conducted, and the development of medical technology, non-invasive respiratory technology, and drugs, so I don't think that it reflects the current situation, but what do you think.

Answer

We thank the reviewer for raising this issue. Indeed, the technical performance of ventilator have improved. However, they are far from being perfect and for instance, leaks (associated with poor sleep quality and lower sleep duration in our study) are still an issue. In the revised version of the manuscript, we have added to the limitation section of the discussion a sentence reporting this issue. It reads as follows: “Second, this study is a post-hoc analysis of a prospective cohort of patients with ARF which primary objective was not to study sleep and that was conducted 10 years ago. Given the improvement of some technologies, it might not reflect the current situation. However, despite the improvement of the performance of ventilators, leaks for instance are still an issue in some patients”.

2)

Reviewer

In data collection, you assessed the quality of sleep with interview using with defined score by nurses  following the first night in the ICU. But Is it acceptable to assess sleep quality only on the first day of ICU admission? I think it's difficult to judge from the quality of sleep on the first day alone, because daily sleep disorders in the ICU may cause delirium. What do you think about it? What is your opinion?

Answer

The reviewer is totally right. It is definitely possible that sleep was different on the second or the third night as compared to the first night. This important limitation of our study is now stated in the discussion as follows: “Fifth, we only measured sleep quality and quantity once, on the first nigh following admission. We cannot exclude that sleep quality and quantity were different on the second or third night as compared to the first night”.

3)

Reviewer

Is it possible to make a proper evaluation by subjectively evaluating the defined factors associated with this interview with patients, because there are significant differences in the values of individual patients? We think that the evaluation with more objective data such as electroencephalography and capnometers is more accurate. What do you think about it?

Answer

We totally agree with the reviewer. In the one hand, the assessment of sleep with objective tools such as EEG is more accurate. However, on the second hand this is less feasible and we could not reach a so high sample size with EEG. This point is acknowledged in the discussion section of the revised as follows: “First, sleep was assessed in a subjective way. Although polysomnography is the reference method, it remains difficult to perform to and interpret properly in the ICU setting. The use of a subjective method was more feasible. Despite a tendency to overestimate quality and duration of sleep, there is a correlation between subjective assessment and polysomnography [38,39]. In addition, PADIS guidelines suggest not routinely using physiologic sleep monitoring in the ICU and to prefer monitoring of patients perceived sleep [16]”.

4)

Reviewer

In result and discussion, you mentioned that air leak and anxiety under NIV were related to quality of sleep with significant difference. Do you have any solution of this problem? Under mechanical ventilation, in general, we use sedation drugs like fentanyl, propofol or others to manage sedation during ventilator management. However, under NIV, I don't think there are clear rules for sedation, have you taken any measures against this? Or do you take any steps to improve quality of in response to this result, for example, using with dexmedetomidine or psychiatric drugs? 

Answer

We thank the reviewer to point this clinical issue. Regarding air leaks, we feel that improvements in interfaces and technical performance of ventilator may in part solve the issue. Regarding anxiety, we feel that the benefit of anxiolytic medications such as benzodiazepines should be evaluated in ICU patients. Indeed, anxiety is an ICU stressor, among others. For instance, pain is cautiously monitored and treated in ICU patients. Why not anxiety? Future studies are needed.

Reviewer 2 Report

Review

1.       As I understand, this was a posthoc analysis of a previous study. Was measuring sleep duration and assessing quality part of the original protocol? If not, how was it standardized across the several institutions and care providers?

2.       Is patient self-reported sleep quality and the scale used previously validated measures of assessing sleep quality?

3.       Please elaborate on what the multivariable models controlled for. The explanation of the multivariable models is slightly confusing. E.g. for NIV failure you mention PaCO2 on admission was associated with NIV success (OR 0.98, 95% CI 0.96-0.99, P=0.003), while short sleep duration (OR 0.77, 95% CI, 0.66-0.88; P=0.001), de novo ARF (OR 3.21, 95CI 1.11-10.6, p=0.039) and SOFA score (OR 1.33, 95% CI, 1.20-1.49, p<0.0001) were associated with NIV failure. Are these per level of PaCO2 or per hour of sleep duration and per unit of SOFA score? Moreover, if the outcome was NIV failure, then sleep duration was actually associated with lower odds. However if it was NIV success then it is the other way around. It would be helpful to explain the multivariable models in more detail, preferable with separate tables.

4.       Page 7, line 227-229, please mention the measure of association for factors associated with short sleep duration.

5.       The major findings are somewhat confusing. Is the sleep quality poor due to the fact that NIV failed and these patients were sicker? The clinical implication of the finding is limited as patients who are not responding to NIV will naturally not sleep well. It does not necessarily mean that poor sleep quality or duration led to NIV failure or improving sleep quality will improve NIV success.

6.       In the discussion, the authors mention that perceived sleep quality was poor with a median score of 2. However earlier they included 2 or more as a marker of acceptable sleep quality. Authors need to clarify what they consider as acceptable sleep quality.

7.       The relationship between sleep and NIV failure section in discussion is confusing. First, the use of term risk of NIV failure is misleading as it was not a risk but an association. Second the authors mention there were abnormal sleep patterns, duration of REM sleep which were not even studied.

8.       The discussion needs to be modified to reflect as to what the authors feel is the clinical implication of this study.

Minor comments

Line 86: Nighànight

Line 87: various causeà varied etiology

Needs grammatical and language review for clarity.

Author Response

We thank the Reviewer for her/his careful reading and positive appraisal of our manuscript and for his comments and suggestions that we have taken into account.

1)

Reviewer

 As I understand, this was a posthoc analysis of a previous study. Was measuring sleep duration and assessing quality part of the original protocol? If not, how was it standardized across the several institutions and care providers?

Answer

Yes, the assessment of sleep duration and quality was part of the parent protocol and these two variables were collected in the eCRF at the initial phase of the study.

2)

Reviewer

Is patient self-reported sleep quality and the scale used previously validated measures of assessing sleep quality?

Answer

No, unfortunately, the scale we used was not validated by previous studies. Scores to evaluate the sleep quality have not been validated in the ICU population. Moreover, these scores (such as the Pittsburg Sleep Quality Index, PSQI) as designed for more stable patients. We would like to point that, in the absence of validated tool, analog numeric scales (which is the type of scale that we used) are not so bad.

3)

Reviewer

Please elaborate on what the multivariable models controlled for. The explanation of the multivariable models is slightly confusing. E.g. for NIV failure you mention PaCO2 on admission was associated with NIV success (OR 0.98, 95% CI 0.96-0.99, P=0.003), while short sleep duration (OR 0.77, 95% CI, 0.66-0.88; P=0.001), de novo ARF (OR 3.21, 95CI 1.11-10.6, p=0.039) and SOFA score (OR 1.33, 95% CI, 1.20-1.49, p<0.0001) were associated with NIV failure. Are these per level of PaCO2 or per hour of sleep duration and per unit of SOFA score? Moreover, if the outcome was NIV failure, then sleep duration was actually associated with lower odds. However if it was NIV success then it is the other way around. It would be helpful to explain the multivariable models in more detail, preferable with separate tables.

Answer

We thank the reviewer for her/his comment, which helped us to improve the presentation of our results. First, in the revised version of the manuscript, we have included the two multivariate analyses in the Table 3, as suggested by the reviewer. Second, we have simplified the text in the results section, which now reads: “On multivariate analysis model including perceived sleep quality and after the selection process, three factors independently predicted the result of NIV: PaCO2 on admission was associated with NIV success, while poor sleep and SOFA score were associated with NIV failure. On multivariate analysis model including sleep duration and after the selection process, four factors independently predicted the result of NIV: PaCO2 on admission and sleep duration were associated with NIV success, while de novo ARF and SOFA score were associated with NIV failure”.

4)

Reviewer

The major findings are somewhat confusing. Is the sleep quality poor due to the fact that NIV failed and these patients were sicker? The clinical implication of the finding is limited as patients who are not responding to NIV will naturally not sleep well. It does not necessarily mean that poor sleep quality or duration led to NIV failure or improving sleep quality will improve NIV success.

Answer

Our results do not allow us to establish a causality link between sleep and NIV failure. However, we underline the fact that poor sleep and severity (assessed with the SOFA score) were both independently associated with NIV failure. This suggests strongly that the link between poor sleep and NIV failure is not exclusively explained by the fact that the sickest patients are those who have a poor sleep and who are at high risk of failing NIV. Accordingly, we have added the following sentence to the discussion: “Because this independent association was adjuted on severity (SOFA score), it suggests that the link between poor sleep and NIV failure is not exclusively explained by the fact that the sickest patients are both at a higher risk of poor sleep and NIV failure”.

Whether improving sleep quality and quantity improves NIV failure is another question, at which this study cannot answer.

5)

Reviewer

In the discussion, the authors mention that perceived sleep quality was poor with a median score of 2. However earlier they included 2 or more as a marker of acceptable sleep quality. Authors need to clarify what they consider as acceptable sleep quality.

Answer

We agree with the reviewer that this sentence is misleading. In the revised version of the manuscript, we have changed the sentence, which now reads as follows: “The perceived sleep quality was globally low with a median score of 2 on a semi-quantitative scale range from 1 to 4 agreed with previous published data”.

6)

Reviewer

The relationship between sleep and NIV failure section in discussion is confusing. First, the use of term risk of NIV failure is misleading as it was not a risk but an association. Second the authors mention there were abnormal sleep patterns, duration of REM sleep which were not even studied.

Answer

We thank the reviewer for her/his suggestion. In the first sentence of the paragraph, we have replaced the term “risk” by the term “rate”. The sentence now reads: “We found that poor sleep quality and short sleep duration were associated with a higher rate of NIV failure and subsequent intubation”. Second, we have removed the paragraph dealing with sleep pattern.

7)

Reviewer

The discussion needs to be modified to reflect as to what the authors feel is the clinical implication of this study.

Answer

We thank the reviewer for this suggestion. In the revised version of the manuscript, we have expanded the conclusion, which now conveys our main messages regarding the results of our study. It now reads: “In conclusion, sleep of patients with ARF requiring NIV is impaired regardless of the etiology of ARF. Poor sleep is definitely an ICU stressor, among others like anxiety, pain, thirst and dyspnea. Subsequently, poor sleep requires the attention of stakeholders, such as pain. Improving the treatment of poor sleep in order to improve patients comfort desserves an evaluation. Future work may compare various strategies to improve sleep in ICU patients and assess the impact of these strategies on short term outcomes such as NIV failure, but also on middle and long term outcomes such as the ICU burden.

Sleep could also be used as a monitoring tool to anticipate NIV failure and subsequently to avoid delaying to much intubation. Future works might assess the benefit of routine sleep monitoring of patients admitted for ARF.”.

8)

Reviewer

Line 86: Nighànight

Line 87: various causeà varied etiology

Answer

“Causes” have been replaced by “etiology”. We could not find the typo with night.

9)

Reviewer

Needs grammatical and language review for clarity.

Answer

The manuscript has been edited for english by a professional English editor.

Round 2

Reviewer 1 Report

Thank you for giving me the opportunity to check your manuscript again. I totally agree with your answer to our comments. I can understand your opinions and the meaning of this article. I accept your report.

Reviewer 2 Report

-